# Canine Distemper Virus in Endangered Species: Species Jump, Clinical Variations, and Vaccination

**DOI:** 10.3390/pathogens12010057

**Published:** 2022-12-29

**Authors:** Rebecca P. Wilkes

**Affiliations:** Department of Comparative Pathobiology, College of Veterinary Medicine, Purdue University, West Lafayette, IN 47907, USA; rwilkes@purdue.edu

**Keywords:** canine distemper virus, vaccination, endangered species, species jumping

## Abstract

Canine morbillivirus (Canine distemper virus, CDV) is the cause of distemper in a large number of different species, some of which are endangered. The clinical outcome associated with infection is variable and based on many factors, including the host species, the immune response of the individual animal to the infection, and variation in virus tropism and virulence. Unfortunately, the viral characteristics associated with virulence versus attenuation are not fully characterized, nor are the specific mutations that allow this virus to easily move and adapt from one species to another. Due to its wide host range, this virus is difficult to manage in ecosystems that are home to endangered species. Vaccination of the domestic dog, historically considered the reservoir species for this virus, at dog-wildlife interfaces has failed to control virus spread. CDV appears to be maintained by a metareservoir rather than a single species, requiring the need to vaccinate the wildlife species at risk. This is controversial, and there is a lack of a safe, effective vaccine for nondomestic species. This review focuses on topics that are paramount to protecting endangered species from a stochastic event, such as a CDV outbreak, that could lead to extinction.

## 1. Introduction

Canine morbillivirus (Canine distemper virus, CDV) is a member of the family *Paramyxoviridae*, genus *Morbillivirus*, which includes *Measles morbillivirus* in humans and nonhuman primates, *Small ruminant morbillivirus* (*Peste-des-petits-ruminants virus)* in sheep and goats, *Phocine morbillivirus* and *Cetacean morbillivirus* in marine mammals, and *Rinderpest morbillivirus* in artiodactyls (now extinct), among others [1,2,3] (Virus Taxonomy: 2021 Release https://ictv.global/taxonomy, accessed on 16 November 2022). Morbilliviruses are considered to be the most infectious viruses known, and morbidity and mortality rates of 90–95% are not unusual in naïve populations [4].

CDV is an enveloped, non-segmented, single-stranded, negative-sense RNA virus that codes for six structural proteins and two non-structural proteins. The structural proteins carried as part of the virus include the RNA polymerase (L) that is associated with the nucleocapsid protein (N) and a phosphoprotein (P), which serves as a cofactor for the polymerase; the matrix protein (M), which lies between the nucleocapsid and the viral envelope, which contains two remaining structural glycoproteins, the hemagglutinin (H), and the fusion (F) protein [2]. The P gene also encodes two nonstructural proteins, V and C, which function as accessory proteins that regulate the host’s immune response and enhance replication [5,6,7]. Specifically, the V protein is a virulence factor involved in modulation of innate immune responses. The C protein is a polymerase cofactor required for processive viral RNA synthesis, preventing the development of double-strand RNA molecules that activate the innate immune response. The C protein is important for virus virulence, and C deficient CDVs are attenuated in vitro and in vivo [5].

CDV, the cause of the clinical syndrome canine distemper (CD), is thought to have originated from measles virus in a species cross-over event, from humans to domesticated dogs (*Canis lupus familiaris*). The first credible reports of CD in dogs occurred in South America in 1735, following the introduction of the measles virus with the arrival of the Europeans. The measles epidemics in indigenous South American populations that followed likely facilitated measles transmission and adaptation to dogs, and this is supported by how closely CDV is molecularly related to measles virus [4]. Additionally, CDV and measles virus produce similar systemic disorders and have comparable incubation periods in their respective hosts [2,8]. CD was then introduced into Europe, followed by the introduction to the colonies in North America, with the first reports of CD in dogs in North America in the 1760s [4,9]. CDV has a practically global distribution and still remains a significant pathogen for domestic dogs, particularly for unvaccinated populations [8]. The incidence of CDV related disease in canine populations throughout the world seems to have increased in the past decades, and several cases of CD in vaccinated animals have been reported [10].

## 2. Reservoir Hosts and Host Range

Domestic dogs have historically been considered the reservoir for CDV. A carrier state has not been recognized for the virus, so the agent persists by transmission to offspring of non-immune dams or to adult dogs lacking active immunity [9]. Spread of this virus is influenced by the presence of the viral envelope. The envelope makes the virus labile, easily inactivated by detergent alone, and reduces its ability to remain infectious for long periods of time in the environment (lasting only a few hours at room temperature, though longer in cold temperatures during winter months). Thus, the spread of this virus is by direct contact with an infected individual (e.g., mating, fighting, grooming, predation), by exposure to respiratory droplets or aerosolized infected bodily fluids, or less commonly through fomites [10,11,12,13].

Unlike measles virus that is maintained by a single host species, CDV is a promiscuous virus with the ability to infect many host species. The dog is only one of several species of order Carnivora that is at risk for infection [3,8,11]. The first report of CDV in a non-domestic species occurred in 1937 in an outbreak in silver jackals (*Vulpes chama*) in a zoo in Johannesburg, South Africa [14]. This was followed by a report from the US, with the first case of CDV in the American badger (*Taxidea taxus)* in Colorado in 1942 [14]. Felids can also be infected with CDV and cases of CD were first reported in the US in samples from captive exotic felid species infected between 1991 and 1992. However, retrospective testing of samples from large felids from zoos in Switzerland confirmed the presence of CDV antigen in tissues from these species since 1972 [14].

By the 1960s efficacious modified-live virus (MLV) vaccines were developed and had a dramatic impact on the incidence of CD in the domestic dog population. Vaccination of dogs has contributed to a decrease in percentage of CDV-infected dogs in countries where high vaccine coverage exists, which now leaves wild carnivores such as raccoons (*Procyon lotor*) as the main reservoir [15,16]. The immune response to CDV is long-lasting, and potentially life-long [2,8]. Thus, for this virus to be maintained in a population in an enzootic state, the population must be large enough to produce a continuous source of susceptible animals [4]. Raccoons, like dogs, can exist at high densities in urban environments and due to high contact rates of these populations, the risk of raccoons serving as a potential reservoir for CDV infection is great and has been demonstrated, for example, in the US and Germany [15,16,17,18,19]. In fact, raccoons have been considered responsible for the spread of CDV to animals both in the wild and in zoological collections [15,18,20,21]. Reports of increased circulation of CDV in other wildlife species, such as the Eurasian badger (*Meles meles*), stone marten (*Martes foina*), European marten (*Martes martes*), the European polecat (*Mustela putorius*), and red fox (*Vulpes vulpes*) in Europe [22]. Infected wildlife species can actually be a source of CDV infection for domestic dogs [13,19].

The viruses’ RNA polymerase that converts the negative strand genome template into positive strand mRNA for protein coding lacks proofreading ability, resulting in the high mutation rate that is seen with RNA viruses such as CDV [9]. This creates the potential for emergence of antigenic variants, particularly in situations where wildlife are infected with a strain of CDV that may have adapted to that host that then spills back to domestic dogs [23]. Introduction of novel canine distemper viruses in improperly vaccinated dog populations with insufficient immunity can cause new outbreaks of CDV [13]. Therefore, even in areas where the virus is considered enzootic, epizootics continue as the virus moves from one animal population or species to another [24].

As suggested with CDV in domestic dog and raccoon populations, it is a “crowd” disease, requiring interacting populations of animals to maintain the virus in an enzootic state. CDV generally displays a “boom and bust” infection cycle. In nature, a new variant is introduced into a naive population, creating high morbidity and mortality, followed by purifying selection of the virus. In smaller populations lacking a constant supply of susceptible animals, generally regular epizootics are followed by “fade-outs’’ during which infection disappears until reintroduced from outside [4]. Most terrestrial mammals, unlike domestic dogs in dense urban settings, naturally live in social groups that are too small to maintain enzootic (versus epizootic) infections of viruses that cause crowd diseases, and are unlikely hosts for the maintenance of CDV [4]. The need for direct contact with an infected individual has an effect on the viruses’ ability to spread within wild populations, particularly if the animal species is mainly solitary, such as tigers and leopards. Infections in solitary species are generally not by conspecifics but by introduction of the virus by a different species [25]. Spread is potentially through predation. For example, the leopard can prey on various virus hosts (free-ranging domestic dogs and small sized carnivores such as the Eurasian badger and red fox), which may act as infection pools [9,25,26].

Though CDV is commonly reported in carnivores, the host range of the virus continues to expand. To date, CDV has been reported in animals in at least four additional orders, including Rodentia, Artiodactyla, Pilosa, and Primates [11,12,14,27,28]. The range of animals is quite diverse and not typical of most animal viruses, and this demonstrates the evolutionary potential of these highly infectious viruses and makes CDV currently one of the most dangerous members in the morbillivirus family [4,8]. Recurrent epizootics among an increasing number of wild, feral, and domestic animal species has raised concerns about the extinction threat it poses to several endangered wildlife species [1,4,12].

## 3. Threat to Endangered Species

For conservation managers, evaluating the degree of disease threat must address both the likelihood of pathogen introduction into an endangered population and its potential impact on population viability [29]. Many endangered populations are small, isolated, and vulnerable to stochastic processes such as outbreaks of infectious diseases [30]. Highly pathogenic infections, such as CDV, pose an immediate extinction risk to small- and medium-sized populations [31].

Sporadic spillovers of CDV into wildlife has resulted in devastating mortality events with outbreaks confirmed to be related to the declines or near extinction of several wild animal populations [1]. These populations have included the black-footed ferret (*Mustela nigripes*), Santa Catalina Island fox (*Urocyon littoralis catalinae*), African wild dog *(Lycaon pictus*), and Caspian seal (*Pusa caspica*) [9,20,25,32,33]. Outbreaks have also occurred in captive breeding facilities for endangered African wild dogs and threatened giant pandas (*Ailuropoda melanoleuca*) [9,34,35].

CDV is regarded as the pathogen of greatest threat to large felids worldwide, with disease and mortalities reported in many species of genera *Panthera* and *Lynx* [25,26]. Over 85% of the lion (*Panthera leo*) population in the Serengeti National Park in Tanzania was infected in the early 1990s, which led to loss of 30% of the population [9,11,36]. CDV also caused high mortality in the Ngorongoro Crater lion population in 2001 [9]. CDV is a current threat to the Amur leopard (*Panthera pardus orientalis*) [26], the Javan leopard (*Panthera pardus melas*) [30], the Amur tiger (*Panthera tigris altaica*) [25], and the Asiatic lion (*Panthera leo persica*) [37], all of which are endangered subspecies.

## 4. Species-Jumping (Host Range Specificity)

The ability of CDV to easily switch hosts has added to concerns about its extinction threat to several endangered wildlife species [1]. How is CDV such a promiscuous virus? Host range specificity of a virus is determined by mechanisms including the means by which viruses gain entry to host cells via their cellular receptors, the ability of the virus to replicate in the new host, followed by the ability of the host to respond to these viral infections through their innate and/or adaptive immune responses [1,2,6,7].

For virus entry, the host cellular receptors used by CDV are (1) SLAM (signaling lymphocyte activation molecule, CD150) located on B and T cells, dendrites, and macrophages, and (2) nectin-4, also known as poliovirus receptor-like protein-4 (PVRL4), on epithelial cells [2]. Preferential binding to SLAM creates the host specificity for morbilliviruses, for example, measles virus preferentially binds to human SLAM receptors. It is CDV’s ability to bind and efficiently use the SLAM receptor of many different host species that provides its promiscuity [38]. This binding occurs despite some differences in SLAM sequences between different species of animal hosts. Unlike SLAM, Nectin-4 is highly conserved in its amino acid sequence among different mammals and therefore likely only plays a minor role or no role in host specificity [2,38]. There is potentially a third cellular receptor used by CDV to infect astrocytes, but this has not been confirmed [39]. The binding of CDV to host cell receptors not only provides host range specificity, but also tissue tropism, which will be discussed in Section 6.

The virus protein responsible for attachment to these host cell receptors is the hemagglutinin (H) protein on the surface of the viral envelope. The fusion (F) protein of the virus, also located on the viral envelope, then facilitates fusion between the envelope of the virus and the host cell membrane [12]. Considering its surface location and the importance of its interaction with the host cell receptors, as expected, the H protein has the greatest genetic variation of all the six structural proteins described for CDV. Thus, the H gene has been used for phylogenetic analysis of CDV. Genotypes (or genetic lineages) are defined on the basis of strains falling within the same phylogenetic clade and sharing > 95 % amino acid similarity in their H protein [1]. There are 22 genotypes currently described. It appears these genotypes were introduced into different parts of the world and have undergone purifying selection independently but in parallel, resulting in geographic clusters or lineages and have been named based on their original geographic locations at time of discovery. These lineages include America-1 (contains almost all commercially available vaccine strains), America-2, America-3, South America/North America-4, America-5, Canada-1, Canada-1, Asia-1, Asia-2, Asia-3, Asia-4, Asia-5, Asia-6, Caspian, Europe/South America-1, South America-2, South America-3, Europe Wildlife, Arctic-like, Rockborn-like (vaccine strain), Africa-1, Africa-2, and Australia [32,40,41,42].

## 5. H Gene Variability

It is expected that variability in the H gene would occur as a result of infection of and adaptation to a new host. Thus, the H gene has been evaluated for mutations that may have facilitated species-jumping. These mutations are predicted to be nonsynonymous (causing a change in an amino acid) and to be under positive selection pressure [43,44]. Though the vast majority of the codons of CDV, including those of the H protein, are under negative or purifying selection, residues 519, 530 and 549 that are within the SLAM binding region of the H protein are under positive selection [43,44]. Residues 530 and 549 in particular have gained considerable interest because they were proposed to be associated with distemper adaptation in non-dog hosts [1,44]. However, studies have shown instead that site 530 is generally conserved within genetic lineages, regardless of host species [1,45]. A mutation at residue 549 (change from tyrosine (Y) to histidine (H)-Y549H), however, may serve some important function. As mentioned, McCarthy et al. [44] suggested this mutation was associated with jump from dogs to non-dogs, but independent reports have shown that a Y at 549 can be found in CDV from both dog and non-dog hosts [1]. Other research has shown that the Y is more commonly found in CDV from dogs and wild canid hosts, compared to non-canid hosts [1,45]. Non-canid strains showed no significant bias towards either H or Y at site 549, although there was a trend towards 549H [45].

Despite the observed trend of the Y549H mutation with non-canid hosts, there has been no clear association of H at 549 with sequences obtained from felids [21,43]. However, in a recent study evaluating CDV strains from African lions that compared sequences from virulent versus non-virulent strains, Y549H was correlated with the presentation of clinical distemper, suggesting this mutation is associated with virulence (as opposed to species-jumping) of CDV in this species [43]. As suggested by Weckworth et al., the discrepancy between finding Y549H in some but not all non-canid hosts might be explained by the dominant cross-species transmission during an outbreak. The amino acid at site 549 of the H protein may be driven by the species responsible for the outbreak, i.e., the reservoir. Thus, if the reservoir is a canid species, a Y might be expected, but if the reservoir is a non-canid, then an H might be expected [43].

Regardless, considering the occurrence of CDV seroconversion in a wide range of taxa, the ability to bind SLAM does not appear to be the limiting factor in species-jumping [43]. Studies on the impact of specific amino acid substitutions within the H protein are speculative and several other factors could also have contributed to the spread of CDV in new hosts [1]. As has been demonstrated in both experimentally and in natural cross-species infections, no significant species-adaptation among H residues in multi-host strains is needed for CDV species-jump and spread [21,38]. Instead, it appears pleiotropic and temporal, sometimes concurrent, geographical lineages seem to be the cause for CDV evolution rather than new host adaptation [21].

## 6. Virus Tropism and Pathogenesis

There is consistency with the pathogenesis for each animal species for which the clinical disease has been described [32]. Binding of the virus to SLAM provides viral tropism for immune cells and causes viremia and immunosuppression, while binding to nectin-4 provides tropism for epithelial cells and is crucial for the development of clinical signs, cell-to-cell spread, and exit from cells, which results in virus shedding [1,2,12,46]. Nectin-4 has also been suggested to play a role in the neurotropism of CDV; however, other, thus far uncharacterized, receptors might be involved in infection of astrocytes [1,12,47]. These as yet unidentified receptors potentially provide cell-to-cell spread in astrocytes, rather than cytolyic infection [48]. In dogs, the incubation period is 1–4 weeks [3]. Infection of dendritic cells or macrophages occurs prior to spread to local lymph nodes. This results in amplification of virus in B and T cells with an initial viremia and an associated transient fever that peaks 3–6 days after infection. The initial viremia spreads the infection to lymphoid tissues throughout the body and the creation of a generalized immunosuppression due to the lymphocytic tropism of the virus with lymphoid cell apoptosis and altered lymphocyte maturation [3,21]. Amplification of the virus in lymphoid tissues leads to a second viremia with high fever followed by infection of epithelial cells throughout the body, which occurs 6–9 days post-infection. Infection of the epithelial tissues leads to shedding of the virus from all bodily secretions and excretions. As would be expected based on its epithelial tropism, the virus produces respiratory, gastrointestinal, and integumentary disease in all species it is able to infect [2,3,11,12].

Secondary bacterial infections resulting from the viral induced immunosuppression complicate the respiratory and gastrointestinal disease. During the second viremia, the virus can also spread to the central nervous system, leading to acute or persistent neurologic signs [3,10]. Cell types within the CNS that can be infected by CDV include astrocytes, microglia, oligodendrocytes, neurons, ependymal cells, choroid plexus cells and glial cells [3,12]. Lesions in the central nervous system of infected dogs can include any combination of demyelination, neuronal necrosis, gliosis, and nonsuppurative meningoencephalomyelitis [3].

The distribution of lesions either in the white matter or the gray matter appears to be virus strain specific [3]. Acute CNS disease is generally progressive and fatal. Clinical signs are associated with the location of the CNS involvement and may include weakness, paresis, paralysis, circling, head tilt, nystagmus, myoclonus, and seizures. Animals that survive acute CNS disease generally maintain CNS deficits [3,10]. Demyelinating encephalitis occurs with chronic infection and persistence of the virus in the CNS [2,3]. CNS disease may occur in the absence of other clinical signs [11]. Remitting-relapsing disease (called Old Dog Encephalitis) has been described but is exceptionally rare [8,10].

Despite commonality in the pathogenesis of CDV between species, there have been great variations in the clinical disease described, ranging from subclinical infection to death [10,11]. These differences are based on host species differences, individual immune response, and virus virulence [49].

## 7. Host Species Differences

Morbidity and mortality vary greatly between different host species [3]. Some felids are known to seroconvert but do not develop clinical disease, such as domestic cats (*Felis catus*), cougars (*Puma concolor*), and ocelots (*Leopardus pardalis*) [21]. This seems to be the case for most species of bears as well, including the endangered Marsican brown bear (*Ursus arctos marsicanus*), in which the virus has been detected in the absence of distemper-related clinical signs or mortality events [50]. However, there has been a single report of CD in an American black bear (*Ursus americana*) [51]. Several species of mustelids, including the black-footed ferret, Siberian polecat (*Mustela eversmanni*), and the domestic ferret *(Musela putorius*) are extremely susceptible to CDV and have a mortality rate of almost 100% [52]. Other species, such as dogs, have a mortality rate that falls between that of these other species, at approximately 50% [3]. There are multiple factors that might contribute to these variable mortality rates, potentially including virus factors, such as differences in virulence of strains that infected these different species in the various studies. However, as seen with experimental infection of raccoon dogs (*Nyctereutes procyonoides)*, foxes (*Vulpes vulpes)*, and minks (*Neovison vison)* with the same CDV variant, there were differences in the clinical severity and mortality rates between these species (mild form in minks, a moderate form in foxes and a severe disease in raccoon dogs). This confirms that species differences in response to a CDV virus do exist [49]. At least for these three species (raccoon dogs, foxes, and minks), the variations in clinical response were related to the differences in the immune response to the virus by each animal species. This study suggests differences in susceptibility to CDV could be related to distinct host cytokine profiles [49].

## 8. Individual Immune Response

Variations in the immune response to the virus are also seen at the level of the individual animal. Great variations in CD-related clinical parameters, including duration, severity, and clinical symptoms of disease have been described in experimentally or naturally infected animals of the same species [10,49]. Serological confirmation of CDV in healthy animals means that some animals can acquire the infection, which may be subclinical, and remain seropositive for years [14].

As determined in the dog and ferret, both natural hosts for CDV, an intact humoral and cell mediated immune response is necessary for clearance of CDV [3,6]. Antibodies directed against the H and F proteins are important for neutralizing the virus by inhibiting virus binding and fusion, thus preventing infection of host cells, and combined with complement-mediated humoral cytotoxicity, are critical for elimination of virus free particles [3]. In dogs, a detectable virus-specific antibody titer at 10–14 days post-infection (pi) promotes viral elimination [3]. Decreased IFN-γ and IL-4 mRNA responses are evident in animals with fatal disease [49]. A shutdown of the IFN-γ and IL-4 early response (dpi 3 and 7) in ferret and mink species has been associated with diminished CDV-specific virus neutralizing antibodies. Differences in cytokine production with distinct patterns between experimentally infected raccoon dogs, mink, and foxes have been related to disease outcome in these species [49]. Thus, there is a correlation between immunosuppression and virulence, with overwhelming virus infection suppressing the cytokine production that is needed to produce a protective neutralizing response [49].

If the infection is not controlled by the antibody response in dogs, it can spread to the central nervous system. A strong, sustained cytotoxic T cell response independent of the antibody response causes virus elimination, inhibiting persistence of the virus in the CNS and is associated with recovery [3,8]. In general, prompt humoral and cellular antiviral immune reactions are found in recovering dogs, are delayed in persistently infected dogs, and are lacking in animals which die [3]. While it is thought that similar responses occur in other animals, there are many gaps with respect to the immune response in wild species.

## 9. Virus Virulence

Though there are genetically distinguishable CDV variants, so far, only one serotype of CDV is recognized. However, several co-circulating CDV viruses of different virulence and cell tropism have been identified. For example, some strains are associated with a polioencephalitis (e.g., Snyder Hill strain and a strain associated with an epizootic first detected in Switzerland in the spring of 2009 [23]), while others induce a demyelinating leukoencephalomyelitis (DL) (e.g., A75-17 strain) [3]. Why does introduction of one strain result in mass mortalities while introduction of other strains is not related with obvious morbidity or mortality [43]? These differences lie on individual variations among the various strains rather than on particular properties inherent to a given CDV lineage [8,23]. For example, Snyder-Hill is an America-1 strain and the strain from the Switzerland wildlife epizootic in 2009 is a South America/Europe-1 strain [23], but both produced polioencephalitis in the infected host [23]. The A75-17 strain may use a currently undefined receptor on astrocytes that is also potentially used by other strains to produce cell-to-cell spread, with associated evasion of the immune response and persistence of the infection with DL [48]. Additionally, as previously mentioned, the H549Y mutation that is suggested to be a virulence determinant in African lions was found in more than one genetic lineage that caused disease and mortalities in this animal species [43].

Unfortunately, though, the molecular mechanisms differentiating virulent from attenuated strains are poorly understood [7]. To complicate matters, the genetic differences leading to virulence might be different, depending on the host species that is infected [43]. While evaluations of differences between CDV variants have been basically limited to the H gene, there are also differences between variants in other important regions of the virus, such as the catalytic center of the phosphodiester bond formation of the large polymerase protein [43], that potentially result in increased virulence. Virulence inducing mutations have actually been shown to accumulate throughout the genome [6]. Molecular evaluation of other genes in multiple lineages might help elucidate some of these unanswered questions with regard to differences in virus virulence [43]. Certainly other genes to consider are the C/V genes, which as mentioned in the introduction, are involved with virus virulence and immune suppression.

## 10. Management of CDV

For disease control, management directed at the reservoir host is easier than management directed at the spillover host, such as an endangered species [29,31,53]. This is assuming the reservoir host is easily managed. Dogs have historically been considered the reservoir of the virus for wildlife, so mass vaccination campaigns have been used in dog populations bordering wildlife areas in an attempt to limit or inhibit the introduction of the virus to wildlife in both Africa and India [29,31,53,54]. The standard operating procedure to deal with stray/feral dogs in tiger reserves released by the National Tiger Conservation Authority of India in 2020 lists vaccinating dogs against CDV as one of the protocols to be followed to ensure the safety of the tigers from the disease (https://www.downtoearth.org.in/news/wildlife-biodiversity/gir-awaits-locallymade-cdv-vaccine-for-lions-experts-divided-on-outcome-82564, accessed on 2 December 2022). However, these campaigns have been ineffective [55,56,57]. Maintaining the herd immunity to prevent epidemics in the dog populations is not possible for many reasons, including associated costs, difficulties with vaccinating feral or quasi-owned dogs, and misconceptions regarding vaccinations by the dog owners [55]. Additionally, even large-scale vaccination campaigns in dogs surrounding the Serengeti National Park, though having a significant impact on CDV outbreaks in the vaccinated dogs, did not prevent transmission of CDV infection to lions [56]. Rather than this being due to lack of herd immunity in the dogs, it has been proven dogs were not the source of the virus for the lions (see Section 11 below). While it has been suggested these mass vaccination programs for dogs in the dog-wildlife interface areas should be legally mandated [58], it is important to weigh the costs of the expensive programs against the amount of protection that they actually provide [55,56].

## 11. Metareservoir

As previously mentioned, CDV requires a dense population to be maintained in an enzootic state. Based on studies in Africa, it has been determined that dog populations near wildlife areas are not dense enough to produce enough susceptibles to maintain the virus [29,56]. CDV likely burns through a relatively small domestic dog population before enough new susceptibles are born, resulting in epizootics rather than persistence. Newer studies have shown that dogs were not the direct source of CDV for the Serengeti lions in the outbreak in the 1990s, nor have they been the source of CDV for Amur tigers [43,57]. As with dogs, populations of individual wildlife animal species are not large enough to maintain the virus. Conditions that either facilitate or limit the interaction of viruses with large numbers of immunologically naïve hosts are drivers for morbilliviral epizootics and expansion of host ranges for CDV [4]. The ability of species to serve as reservoirs depends on their susceptibility, population size, turnover and frequency of effective contacts [25]. Thus, for persistence to occur in a given ecosystem, there is likely a metareservoir that consists of multiple interconnected carnivore populations comprised of multiple species [25,29]. Therefore, simply controlling or vaccinating free-ranging domestic dogs is not enough to prevent this virus from infecting endangered free-ranging wildlife species [25,26,55,57] Lack of a single reservoir species to direct control efforts results in the need to directly vaccinate the wildlife host of concern [57].

## 12. CDV Vaccine History

A history of CDV vaccines is important to understanding the controversies surrounding the topic of vaccinating wildlife species, because the original MLV vaccines created for CDV viruses that date back to the period from the 1930s to the 1950s are still in use today. The first MLV vaccines fall into the genetic lineage called America-1. The Onderstepoort strain, which is the prototype America-1 lineage, was developed in 1939 from a CDV isolate from North American ranched foxes. The strain was originally serial passed in the ferret but then further attenuated by serial passage in embryonated hen’s eggs. Since the 1950s, this egg-attenuated strain has dominated the market and is contained in most of the currently available vaccines [59,60]. Included among these Onderstepoort vaccines is the vaccine originally known as Galaxy D, which was a stand-alone CDV product marketed by Schering-Plough Animal Health. While the Onderstepoort strain has been previously attenuated in eggs, the Galaxy D vaccine was attenuated by passage through an immortal primate cell line to improve product uniformity [61]. It was shown to be safe and efficacious following a challenge study in domestic ferrets and was marketed for that species. However, Galaxy D is no longer available. This vaccine virus is now part of the Nobivac line of vaccines marketed by Merck Animal Health following their buy-out of Schering-Plough and is marketed as a multivalent vaccine containing additional MLV viruses for dogs [62].

Another MLV vaccine of note is the Rockborn strain. Following a CDV epidemic in Sweden in the 1950s, a canine isolate- Rockborn- was attenuated in primary dog kidney cells and distributed globally as a vaccine after 1962 [59]. However, this vaccine strain has been associated with suspected cases of post-vaccinal encephalitis in dogs and the strain was withdrawn from several markets after the mid-1990s [59]. This vaccine is currently the only one commercially available that is not an America-1 strain, but instead forms its own unique clade (Rockborn-like) [59]. There is a vaccine still commercially available that contains a Rockborn-like variant [63]. Canine cell origin CDV vaccines are hazardous and many of the cases of vaccine induced CDV in nondomestic species were caused by the Rockborn strain vaccine [64], including cases in the endangered red panda (Ailurus fulgens) [9,65] and recently a fennec fox [66]. Any vaccine containing a Rockborn-like strain should not be used in a nondomestic species.

A different type of CDV vaccine that is commercially available is a canarypox virus recombinant vaccine that expresses the CDV H and F proteins of the Onderstepoort strain. The benefit of using this type of vaccine is that it is not a CDV virus and therefore cannot produce CD. The canarypox virus cannot replicate in mammalian cells, so this vaccine is considered very safe for use [67]. This vaccine has been approved for use in ferrets (as a standalone CDV product with the trade name Purevax) and in combination with MLV components for other canine pathogens as multivalent vaccines for dogs. The canine line of this vaccine is Recombitek. Of note, the Recombitek product has fewer plaque-forming units of the recombinant canarypox virus and consequently generates a muted antigenic response in some species compared to Purevax [68].

## 13. Vaccine Use for Nondomestic Species

Vaccination has been very effective to control a virus such as rabies in endangered populations including African wild dogs and in preventing extinction of Ethiopian wolves (*Canis simensis*) [26]. For CDV, though, vaccination of the endangered species has been very controversial [57]. Unfortunately, killed vaccines while safe, are ineffective for CDV [69]. Historically, CDV MLV vaccines have induced disease and caused mortality in several species, including African hunting dog, black-footed ferret, red panda, European mink (*Mustela lutreola*), and kinkajou (*Potos flavus*) [33,70]. There is a general lack of quantitative data on the effect of CDV vaccines in wildlife [1]. Thus, there are questions with regard to the best vaccine to use in these species.

All CDV vaccine use in nondomestic animals is extra-label [11]. Therefore, in the United States, use of vaccines in nondomestic species falls under The Animal Medicinal Drug Use Clarification Act of 1994 (AMDUCA), which requires use of the vaccine within the context of a veterinarian-client-patient relationship (https://www.fda.gov/animal-veterinary/guidance-regulations/animal-medicinal-drug-use-clarification-act-1994-amduca, accessed on 28 November 2022).

Additionally, there are operational difficulties associated with vaccination of wildlife. This requires the availability of funds and potentially enhanced trans-border cooperation. The cost of vaccinating tigers was estimated by Gilbert et al. to be nearly $15,000 per tiger per year [26,57].

At least in some situations, vaccination of wildlife is an accepted practice. This is particularly common for animals that have been trapped for use in breeding programs to be later reintroduced into the wild. For example, in the US, vaccination of black-footed ferrets [71] and Channel Island foxes has been performed as part of recovery efforts for these species (https://www.biologicaldiversity.org/news/press_releases/2016/island-fox-08-11-2016.html, 20 November 2022).

Nondomestic animals in zoo collections have also been vaccinated against CDV. The challenges associated with vaccination of nondomestic species is evident through this work. The vaccines used and the frequency of administration has varied [21,72,73,74]. However, studies in captive animals can help direct management plans and the potential for vaccinating endangered free-ranging species [72]. Challenges that need to be considered include (1) knowledge on the safety and efficacy of the vaccine in the specific species targeted; (2) mode and timing of vaccine delivery; (3) the logistics of administering the required booster shots; (4) the costs associated with a vaccination program in wildlife; and (5) availability of the vaccine [1].

After Purevax for ferrets was licensed and marketed in 2001, many North American zoological institutions began using this vaccine to vaccinate numerous at-risk species. Due to its safety, The American Association of Zoo Veterinarians’ Distemper Vaccine subcommittee recommended the extra-label use of the Purevax Ferret Distemper Vaccine in all susceptible exotic carnivore species [13,75]. However, the vaccine has not been adequately evaluated in all the species for which it has been used. Additionally, there have been Purevax manufacturing issues with associated periods of shortages or lack of availability for extended periods of time in the US [71]. This vaccine is not available in several countries, including those in Latin America [69]. There are also restrictions on the import of genetically modified organisms into many African countries [64], and the recombinant canarypox-vectored vaccines are not available in Japan [66]. Due to the limited supply of the canarypox-vectored CDV vaccine and the potential risks associated with MLV vaccines, many animals in zoos and in breeding programs have not been routinely vaccinated [35,74,75,76]. Some zoos have used outdated Purevax vaccine in their animals, but this may have been less effective [76].

Thus, the Recombitek line for dogs was recommended in place of the Purevax vaccine because it is antigenically similar [68]. Though again, studies to validate its use in nondomestic species had not been performed. Unfortunately, combination vaccines, including the Recombitek line, contain MLV components, presenting additional safety concerns in these species [73]. Use of dog multivalent vaccines has been discouraged because of possible immunosuppression and clinical disease brought about by other MLV components, such as parvovirus [11,64,73].

## 14. Evaluation of Canarypox Recombinant Vaccine in Non-Domestic Species

In recent years, there has been renewed interest in evaluating the effectiveness of the recombinant canarypox vectored CDV vaccine in multiple nondomestic species. Considering these species are endangered and part of breeding/conservation programs, challenge studies for evaluation of vaccine efficiency are not possible. Thus, as is conducted in dog studies, serum neutralization (SN) titers were used as a stand-in to represent possible protection from CDV following vaccination [68]. At least in natural CDV infections in dogs, the magnitude of the SN titer has been correlated with outcome of the infection [3]. However, at least for this particular vaccine, use of SN titer magnitude to suggest protection has been questioned, because a previous study in dogs demonstrated protection provided by the recombinant canarypox vectored vaccine with experimental challenge, despite low SN titers in the dogs [67]. The thought is that the cell-mediated immune response driven by this vaccine is important to the protection it provides, which is not captured by the SN titer that only evaluates the humoral response. Therefore, the measured serologic response alone with this type of vaccine may not adequately assess the degree of protection conferred by vaccination [68,77].

Despite this, based on evaluation of the recombinant vaccine, either as Purevax in giant pandas [75], African wild dogs [72], red pandas [76], meerkats (*Suricata suricatta*) and fennec foxes *(Vulpes zerda)* [77], black-footed ferrets [71] and large felids [73] or as Recombitek in captive tigers [68], maned wolves [78], black-footed ferrets [71], African wild dogs [79], red foxes [80] and red pandas [76], it is evident that there is significant individual animal and species variation with response to these vaccines. At least 2–3 booster vaccines are needed to provide an adequate (or even detectable) humoral response for all species, except potentially Purevax in black-footed ferrets which produced what was considered protective titers with a single dose in a small number of wild ferrets [71]. Annual boosters are recommended for the recombinant canarypox vaccines because titers are short-lived [71,75,80]. In some animals, minimal or no SN titers are produced, even with multiple booster vaccinations. For example, in an evaluation of 159 large felids vaccinated with Purevax, only 66% of the vaccinates seroconverted at some point post-vaccination [73]. In a study in maned wolves, seroconversion was not obtained during the study period, despite booster doses of Recombitek [78].

It has been suggested that low titers may be due to a dose-dependent response to the vaccine in larger animals [81]. This is supported by a challenge study that was performed with the recombinant cararypox vaccine in Siberian polecats, the closest living relative of the black-footed ferret. Survival rate following challenge was 50–60% with 10^5^–10^5.5^ plaque forming units (pfu)/dose, but 100% mortality was seen with a dose of 10^4.5^ pfu [52]. An alternate dose could be considered for larger species [78,81]. However, in a study performed in a limited number of *Panthera* spp., even with use of 3 mL of Recombitek as a booster dose (1 mL is the labeled dose), tigers produced very minimal to no humoral response [68].

Proof that a single dose of Recombitek is not protective is seen from reports of natural CDV exposure and death in maned wolves [69] and in a snow leopard [74], despite recent vaccination with this vaccine. Titers from these animals showed a minimal humoral response to the vaccine, suggesting titers may actually be useful as an indicator for protection for the recombinant vaccine, at least in some nondomestic species [69].

## 15. Vaccination Schedule for Nondomestic Species

Other than the vaccine to be used, another consideration for vaccinating free-ranging wildlife is the vaccination schedule. Vaccination schedules for CDV recommended for nondomestic species are generally based on recommendations for the domestic dog [11]. Recommendations for dogs are vaccinate every 3–4 weeks between 6–16 weeks of age and another booster given at 1 year [11,82]. The boosters are recommended to overcome interference of maternal antibodies, but this period of interference is different in each species. For example, in raccoons, the vaccine should be administered again at 18–20 weeks of age, but in ferrets, the last dose after 10 weeks of age is fine, because of differences in maternal antibody half-life [11]. Again, species differences need to be considered when implementing a plan.

## 16. Use of Booster Vaccinations in Wildlife

The concern with using a vaccine that requires boosters is, how is this going to be properly managed for solitary, reclusive animals, such as leopards and tigers? Should animals be captured for vaccination? As seen with the Asiatic Gir lions following the CDV outbreak in 2018, lions were captured in 2019 for vaccination with Purevax. Due to the need for multiple boosters, these lions were maintained in captivity. The animals are now no longer considered wild and potentially cannot be reintroduced into the wild. As of April 2022, these lions have not been released (https://www.downtoearth.org.in/news/wildlife-biodiversity/gir-awaits-locallymade-cdv-vaccine-for-lions-experts-divided-on-outcome-82564, accessed on 28 November 2022).

Ideally any vaccine that is used for free-ranging animals would not require a booster if administered past the point of potential maternal antibody interference. Additionally, protection afforded by a single dose is important if the use of low-coverage vaccination for development of a core protected group is followed as has been recommended [26,57]. This would involve vaccinating small numbers of animals (e.g., Amur tigers) each year to reduce the risk of extinction. According to Gilbert et al., annual vaccination of two tigers per year would reduce the 50-y extinction probability of the Leopard Land National Park (Russian Far East) population from 15.8% to 5.7% [57].

## 17. MLV Vaccine in Nondomestic Species

At this point, the only potential vaccine available that may work as a single dose is a MLV vaccine [57]. There has been a resurgence in evaluating MLV vaccines in nondomestic species despite concerns of reversion to virulence with vaccination. As described, MLV vaccines have been used very effectively in domestic ferrets, which are highly susceptible to CD. Unfortunately, the safe CDV-only MLV vaccines that were approved for ferrets are no longer available, but the American Ferret Association (AFA) has recommended that when a vaccine licensed against CD in ferrets is unavailable, Merck’s Nobivac DPv vaccine can be used. This vaccine has been used extensively in Europe for protecting ferrets against CDV, and as previously mentioned, Nobivac vaccines contain the CDV strain that was originally marketed as Galaxy D (https://www.ferret.org/pdfs/health/TiterLetter.pdf, accessed on 2 December 2022). Nobivac vaccines have also been used safely in large felid species [21,57,68,73]. Based on a study in a limited number of captive *Panthera* spp., Nobivac CDV vaccine produced more consistent and higher titers in this species than the recombinant canarypox vaccine. Protective titers were evident after a single dose [68]. Geroff et al. showed a low complication rate with use of MLV vaccines in large felids and suggested they may not be as sensitive to adverse effects of MLV CDV vaccines as other exotic carnivores [73]. MLV vaccines have been found to carry a low mortality risk for African wild dogs, too [64].

## 18. MLV Vaccine Shedding 

An additional concern outside of reversion to virulence with use of a MLV vaccine is the potential of shedding of the vaccine virus into the ecosystem. At least some species, such as Siberian polecat x black-footed ferret hybrids and grey foxes shed vaccine virus [52]. Thus, ideally, any vaccine used in any species that may shed vaccine virus (including domestic dogs at the wildlife interface) should be considered safe for use in nondomestic species in the event the vaccine virus comes into contact with an endangered species.

## 19. MLV Vaccine Duration of Protection

Another consideration is duration of protection afforded by a MLV vaccine. Though annual boosters are suggested for the canarypox vaccine, length of protection for MLV vaccines is generally considered substantial. At least for dogs, vaccination every three years, rather than annually, is suggested [82]. A study performed in the endangered red wolf (*Canis rufus*) showed detection of IgG at 3 years post-vaccination with a MLV vaccine (Galaxy D) [70]. Protective titers were maintained through 800 days following vaccination with a MLV vaccine in black-footed ferret x Siberian polecat hybrids [71]. Extended protection is another benefit that would be useful in a vaccination program for free ranging species.

## 20. Route of Vaccine Delivery

Route of delivery should also be considered. MLV vaccines and the recombinant canarypox vaccine have been used IM and could be administered by dart [57]. Oral vaccination, as is performed for Rabies virus with baits, would be a useful method for delivery in free-ranging species [57]. Oral delivery with the canarypox vectored vaccine has been attempted in some species. While it was successful in producing SN titers in Siberian polecats when used at very high pfu/dose, when used in the Purevax formulation, was unsuccessful when administered orally in African wild dogs [72] and in tigers [81].

## 21. Development of a New Vaccine

With increasing reports of CDV epizootics, continuous drivers for CDV evolution, and reports of CD in fully vaccinated domestic dogs, there have been suggestions that the current vaccines may not be providing adequate protection [10,83]. The CDV H protein from currently circulating strains differs by approximately 10% from the available vaccine strains [10]. While there are differences in neutralization between the genetic variants and between currently circulating strains and vaccine strains [83,84], the impact of these differences is difficult to appreciate in well-vaccinated domestic dog populations. As previously mentioned, a specific neutralizing response to the H and F proteins is important to limit virus spread in the host. The H protein is the most important glycoprotein to which the response is directed, followed by the F protein. There are neutralizing epitopes associated with the H protein that are not conserved between variants [84]. Research has shown that when the neutralizing response is limited to the H protein only, it can be inefficient to prevent infection and prevention of clinical signs against a heterologous virus. The added neutralizing response to the more conserved F protein is important to provide the serologic cross-reactivity needed for protection [85]. Although it is unlikely that such antigenic variations may affect the protection induced by vaccine immunization, it is possible that critical amino acid substitutions in key epitopes of the H protein may allow escape from the limited antibody repertoire of maternal origin of young unvaccinated animals, increasing the risk for infection by field CDV strains [10]. Additionally, these variations between strains may become more relevant in a population with reduced vaccination/protection or if recombinant vaccines are used that do not address these strain differences [83,85]. Vaccination is actually driving changes in these viruses, and recombination events between America-1 vaccine strains and wild-type strains have been identified. This suggests that CDV vaccination might also play an important role in shaping virus evolution [85]. Knowledge of the genetic lineages in an ecosystem may become even more important over time if decreases in protection afforded by the available vaccines become more significant.

Regardless of any potential need to develop a more contemporary vaccine, there is still a need for an improved vaccine that is specific for nondomestic species, given all the challenges with the available products, as already mentioned. To date, there has been no interest in developing a commercial vaccine in the US for use in nondomestic species. Of note, in response to an outbreak of CDV in the Asiatic lion population, The Gujarat Biotechnology Research Centre has developed a vaccine for this population in Gir National Park. This vaccine has been made with the genetic lineage detected from an outbreak in these lions in 2018, which was named Asia-5 [37] (https://www.downtoearth.org.in/news/wildlife-biodiversity/gir-awaits-locallymade-cdv-vaccine-for-lions-experts-divided-on-outcome-82564, accessed on 28 November 2022). This is the first time in India that any state government or its department is working on a vaccine for wildlife. The vaccine is currently still being tested with plans to manufacture it for use in the lions. This will be the first CDV vaccine in the world for a Felidae family (https://www.hindustantimes.com/india-news/vaccine-to-stop-killer-cdv-among-lions-in-works-101647542607981.html, accessed on 2 December 2022).

There have been recent experimental vaccines produced using other vectors, such as a replication-defective human adenovirus that expresses the H protein of CDV. This experimental vaccine was shown to produce a humoral immune response in foxes (mean VN titer of 1:38 at 4 weeks post-vaccination), and provided protection against a lethal challenge four weeks after a single intramuscular injection with a very high dose (10^9^ green fluorescent units of the vaccine virus) [86]. This vectored vaccine is bivalent- also expresses a protein of rabies virus, but this vaccine is missing the more conserved F protein of CDV that has been demonstrated to elicit a neutralizing response against more divergent CDV strains [85]. The rabies virus has also been used as a recombinant vector for CDV. Two injections of an inactivated rabies virus expressing CDV H and F proteins produced low-level neutralizing antibodies in ferrets but provided protection against lethal challenge five weeks after the last vaccination in 4/4 animals. In animals that received a single vaccine dose, 3/4 animals survived challenge [85]. Including rabies and CDV in a bivalent vaccine would be quite useful for wildlife vaccination programs, but both of these experimental vaccines have limitations that exclude them as ideal for wildlife use. Controlled attenuation of CDV strains for use as MLV vaccines is potentially a possibility, with manipulation of the RNA-dependent RNA polymerase (L) protein already having been proven to produce attenuation of the virus [87].

## 22. Conclusions

CDV is a highly contagious, promiscuous virus, with the ability to easily jump between different species in multiple orders. Many species that can be severely affected by CDV include those that are endangered on the International Union for Conservation of Nature (IUCN) Red List of Threatened Species. This includes several species of felids that remain in small, isolated numbers that are threatened with extinction by stochastic events, such as a CDV outbreak. As various species can maintain the virus in the environment, likely functioning together as a metareservoir, this makes the control of the virus at the level of the reservoir impossible. As has been proven in multiple ecological settings, vaccination of domestic dogs (historically considered the reservoir of CDV for wildlife species) does not prevent spillover of CDV to these threatened species. Thus, to protect these endangered species, vaccination of the threatened species appears to be a necessary part of a conservation plan.

Unfortunately, there are currently no CDV vaccines that have been produced specifically for these species. This results in reliance on commercially available products that are not ideal for use in nondomestic animals. Based on limited available data, one must weigh vaccine safety against efficacy when considering the two products that have been recommended for the vaccination of large felids. Additionally, cost, availability, and vaccination protocol all have to be considered. These are not decisions to be taken lightly and are unfortunately complicated by the currently limited vaccine options. Ideally, individuals knowledgeable in the multiple aspects of this situation would come together to produce a vaccine product that is safe, provides substantial, consistent and extended protection with a single administration, can be administered by dart (or potentially oral bait), and produces prolonged protection for the individual animal. Such a product would have a significant impact on protecting endangered species from CDV.

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
