# Peer review of "Canine Distemper Virus in Endangered Species: Species Jump, Clinical Variations, and Vaccination"

_pathogens, 2022, doi:10.3390/pathogens12010057_

Round 1

Reviewer 1 Report

The paper by  Rebecca P. Wilkes presents an updated review of the role of CAnine distemper virus in wild life conservation. The paper is well structurated and has a high potential for citations because this information is still needed. Minor changes are needed for publication.

Page 4. First paragraph: Refernce (1) seem not to be the most suitable reference.

Page 4. 2nd Paragraph: "There is potentially a third cellular receptor used by CDV, but this has not been confirmed". mention the kinf of cell that this potential receptor has been found and why it could play a minor or none role host specificity. Perhaps in specific pathogenesis.

Page 5. Item 6. Virus tropism and pathogenesis: In this most of the paragraph author are explaing CDV in dogs pathogenesis, however, some aspects are related to other species. Please revise and apecify in the text to wich spoecies are specific each comment.

Page 6. Item 8. Individual immune response. Please specify that this information ismostly related to dog, and there are multiple gaps in the understanding of other wild species immune response.

Page 7. Molecular evaluation of other genes in multiple lineages might help elucidate some of these unanswered questions. Please include some information about the role of C/V genes in the Virulence/Pathogenesis of the CDV

Page 7. Do not use Cap Letters in the phrase: "procedure to Deal with Stray/Feral Dogs in Tiger Reserves  released by the National Tiger Conservation Authority of India in 2020

Page 11. Item 18. MLV vaccine shedding. Discuss the possible role of MLV vaccine shedding in mutation accumulation as has been set for other viruses (i.e. Polio)

Page 12. Item21. Development of a new vaccine. This item need to be contextualized to other missing papers using updated or new viral vectors such as:  10.1016/j.vetmic.2020.108920

10.3389/fmicb.2020.01070 - 10.1128/JVI.02077-16

Hyperlinks must be presented in the references according to the Journal style.

Nice Work!!

Author Response

Page 4. First paragraph: Refernce (1) seem not to be the most suitable reference.

Additional references were added.

Page 4. 2nd Paragraph: "There is potentially a third cellular receptor used by CDV, but this has not been confirmed". mention the kinf of cell that this potential receptor has been found and why it could play a minor or none role host specificity. Perhaps in specific pathogenesis.

This receptor is thought to be located on astrocytes. This info was added. Some info was also added to the virus tropism and pathogenesis section regarding this receptor. See also virus virulence section.

Page 5. Item 6. Virus tropism and pathogenesis: In this most of the paragraph author are explaing CDV in dogs pathogenesis, however, some aspects are related to other species. Please revise and apecify in the text to wich spoecies are specific each comment.

It is thought that the pathogenesis in all animals that develop CD is the same. See first sentence of the first paragraph. I changed the first sentence some.

Page 6. Item 8. Individual immune response. Please specify that this information ismostly related to dog, and there are multiple gaps in the understanding of other wild species immune response.

Information has been added.

Page 7. Molecular evaluation of other genes in multiple lineages might help elucidate some of these unanswered questions. Please include some information about the role of C/V genes in the Virulence/Pathogenesis of the CDV

This info is briefly mentioned in the intro. I added a sentence to allude to this.

Page 7. Do not use Cap Letters in the phrase: "procedure to Deal with Stray/Feral Dogs in Tiger Reserves  released by the National Tiger Conservation Authority of India in 2020

Corrected.

Page 11. Item 18. MLV vaccine shedding. Discuss the possible role of MLV vaccine shedding in mutation accumulation as has been set for other viruses (i.e. Polio)

This is mentioned in the section- development of a new vaccine

Page 12. Item21. Development of a new vaccine. This item need to be contextualized to other missing papers using updated or new viral vectors such as:  10.1016/j.vetmic.2020.108920

10.3389/fmicb.2020.01070 - 10.1128/JVI.02077-16

Thanks for the reminder. I actually meant to include some info on new experimental vaccines and forgot. See added info.

Hyperlinks must be presented in the references according to the Journal style.

I think the editor is working on this part?

Nice Work!!

Thanks so much!

Reviewer 2 Report

While there have been several reviews focused on aspects of canine distemper virus, this paper is a welcome and novel addition. The author provides a comprehensive and up to date synthesis of CDV ecology, virology and pathogenesis focused on all susceptible hosts (not just dogs which have been well covered in previous reviews). The second half of the manuscript is particularly useful and provides a thorough and clear treatment of the issues and needs for CDV vaccines in zoological collections and free-ranging carnivores that has not been covered in this detail in any previous manuscript.   I only have minor suggestions and comments:  

Section 2. Reservoir hosts and host range

- “...so the agent persists by transmission to offspring of non-immune dams or to adult dogs lacking active immunity…” - why focus on dogs? …based on the abstract I’m assuming that the authors will be emphasizing that many species are able to act as competent hosts. - “(e.g., mating, fighting, grooming)” - predation is thought to be an important means of transmission for large carnivores and might be worth adding here. - Typo: “stone marten (Marten foina)” - should read "Martes foina” - Minor point: "For example, the leopard can prey on various virus hosts (free-ranging domestic dogs and small sized carnivores such as the Eurasian badger, red fox, and leopard cat (Prionailurus bengalensis), which may act as infection pools [9, 25, 26].” - There is some question over whether Prionailurus spp. are competent hosts (see Ohishi et al 2014). Prionailurus spp. carry a threonine at aa position 76 in the binding domain of SLAM, which makes them more similar to domestic cats (which are not susceptible) compared to Panthera spp. (which are susceptible). I would replace leopard cat with a species where susceptibility is undeniable (raccoon dog, or sable) - “To date, CDV has been reported in animals in four additional orders, including Rodentia, Artiodactyla, Proboscidea, and Primates” - Oni et al 2006 based their elephant report on serological findings. Considering that several morbilliviruses are antigenically cross-reactive, it is possible that the elephants had been infected with another morbillivirus (previously described or otherwise). How about replacing elephants with sloths (Xenarthra, Watson et al 2020)?   Section 3. Threat to endangered species  - Very minor point: “...all of which are endangered species.” - strictly speaking these are subspecies (although leopards P. pardus are globally vulnerable at the species level).   Section 5. H gene variability - Typo: “Thus, if the reservoir is a canid species, a T might be expected, but if the reservoir is a non-canid, then an H might be expected “ - should read: “Thus, if the reservoir is a canid species, a Y might be expected, but if the reservoir is a non-canid, then an H might be expected “   Section 6. Virus tropism and pathogenesis  - Could you add a citation for “In dogs, the incubation period is 1-4 weeks ”?   Section 7. Host species differences. - "This seems to be the case for most species of bears as well, including the endangered Marsican brown bear (Ursus arctos marsicanus), in which the virus has been detected in the absence of distemper-related clinical signs or mortality events [47]” - Agreed that clinical disease is at most exceptionally rare in bears, but note Cottrell et al 2013 describing clinical disease in an American black bear.   Section 10. Management of CDV - “For disease control, management directed at the reservoir host is easier than management directed at the spillover host, such as an endangered species “ - This is only really true if your reservoir is in dogs! …if your reservoir is in a community of wild mesocarnivores then management within the reservoir is extremely difficult (logistically/financially impossible). - “Additionally, even large-scale vaccination campaigns in dogs surrounding the Serengeti National Park, though having a significant impact on CDV outbreaks in the vaccinated dogs, did not prevent transmission of CDV infection to lions [52]. “ - It should probably be made clear that this is thought because there is a wildlife reservoir in the area, rather than insufficient vaccination of the dogs.   Section 11. Metareservoirs - “As with dogs, each wildlife animal species within itself is not large enough to maintain the virus. “ - suggest minor rewording to “As with dogs, populations of individual wildlife animal species are not large enough to maintain the virus. “   Section 13. Vaccine use for non-domestic species - “Historically, CDV MLV vaccines have induced disease and caused mortality in sev-eral species, including African hunting dog, black-footed ferret, red panda, European mink (Mustela lutreola), and kinkajou (Potos flavus) [31, 66]. “ - I understand that none of these vaccine-related mortalities was based on Onderstepoort strains; is that supported in the literature? …if it is then it would be useful to state that clearly here. - “Therefore, use of vaccines in nondomestic species falls under The Animal Medicinal Drug Use Clarification Act of 1994 ” - should read ““Therefore, in the United States use of vaccines in nondomestic species falls under The Animal Medicinal Drug Use Clarification Act of 1994"

Author Response

So sorry I forgot to add line numbers to the text to make this easier.

  • “...so the agent persists by transmission to offspring of non-immune dams or to adult dogs lacking active immunity…” - why focus on dogs? …based on the abstract I’m assuming that the authors will be emphasizing that many species are able to act as competent hosts. That is correct Yes, this is correct
  • - “(e.g., mating, fighting, grooming)” - predation is thought to be an important means of transmission for large carnivores and might be worth adding here. - Added.
  • Typo: “stone marten (Marten foina)” - should read "Martes foina” - Corrected.
  • Minor point: "For example, the leopard can prey on various virus hosts (free-ranging domestic dogs and small sized carnivores such as the Eurasian badger, red fox, and leopard cat (Prionailurus bengalensis), which may act as infection pools [9, 25, 26].” - There is some question over whether Prionailurus spp. are competent hosts (see Ohishi et al 2014). Prionailurus spp. carry a threonine at aa position 76 in the binding domain of SLAM, which makes them more similar to domestic cats (which are not susceptible) compared to Panthera spp. (which are susceptible). I would replace leopard cat with a species where susceptibility is undeniable (raccoon dog, or sable) - removed leopard cat
  • To date, CDV has been reported in animals in four additional orders, including Rodentia, Artiodactyla, Proboscidea, and Primates” - Oni et al 2006 based their elephant report on serological findings. Considering that several morbilliviruses are antigenically cross-reactive, it is possible that the elephants had been infected with another morbillivirus (previously described or otherwise). How about replacing elephants with sloths (Xenarthra, Watson et al 2020)?  Good point! I just copied what all others have done without really thinking about this. Poor form on my part. I removed the order with elephants and added Pilosa.
  • Section 3. Threat to endangered species  - Very minor point: “...all of which are endangered species.” - strictly speaking these are subspecies (although leopards P. pardus are globally vulnerable at the species level). - Thanks. Corrected to subspecies.  
  • Section 5. H gene variability - Typo: “Thus, if the reservoir is a canid species, a T might be expected, but if the reservoir is a non-canid, then an H might be expected “ - should read: “Thus, if the reservoir is a canid species, a Y might be expected, but if the reservoir is a non-canid, then an H might be expected “  Oops, thanks for catching this!
  • Section 6. Virus tropism and pathogenesis  - Could you add a citation for “In dogs, the incubation period is 1-4 weeks ”?  Added
  • Section 7. Host species differences. - "This seems to be the case for most species of bears as well, including the endangered Marsican brown bear (Ursus arctos marsicanus), in which the virus has been detected in the absence of distemper-related clinical signs or mortality events [47]” - Agreed that clinical disease is at most exceptionally rare in bears, but note Cottrell et al 2013 describing clinical disease in an American black bear.  Added
  • Section 10. Management of CDV - “For disease control, management directed at the reservoir host is easier than management directed at the spillover host, such as an endangered species “ - This is only really true if your reservoir is in dogs! …if your reservoir is in a community of wild mesocarnivores then management within the reservoir is extremely difficult (logistically/financially impossible). Added a sentence to address this.
  • - “Additionally, even large-scale vaccination campaigns in dogs surrounding the Serengeti National Park, though having a significant impact on CDV outbreaks in the vaccinated dogs, did not prevent transmission of CDV infection to lions [52]. “ - It should probably be made clear that this is thought because there is a wildlife reservoir in the area, rather than insufficient vaccination of the dogs.  Added a sentence to address this.
  • Section 11. Metareservoirs - “As with dogs, each wildlife animal species within itself is not large enough to maintain the virus. “ - suggest minor rewording to “As with dogs, populations of individual wildlife animal species are not large enough to maintain the virus. “  Changed
  • Section 13. Vaccine use for non-domestic species - “Historically, CDV MLV vaccines have induced disease and caused mortality in sev-eral species, including African hunting dog, black-footed ferret, red panda, European mink (Mustela lutreola), and kinkajou (Potos flavus) [31, 66]. “ - I understand that none of these vaccine-related mortalities was based on Onderstepoort strains; is that supported in the literature? …if it is then it would be useful to state that clearly here. - Unfortunately, it is not clear in the literature which particular vaccines were used in all these cases.
  • “Therefore, use of vaccines in nondomestic species falls under The Animal Medicinal Drug Use Clarification Act of 1994 ” - should read ““Therefore, in the United States use of vaccines in nondomestic species falls under The Animal Medicinal Drug Use Clarification Act of 1994" Corrected